# PROJECTED SUBNETWORKS SCALE ADAPTATION

## ABSTRACT

Large models support great zero-shot and few-shot capabilities. However, updating these models on new tasks can break performance on previous seen tasks and their zero/few-shot unseen tasks. Our work explores how to update zero/few-shot learners such that they can maintain performance on seen/unseen tasks of previous tasks as well as new tasks. By manipulating the parameter updates of a gradient-based meta learner as the projected task-specific subnetworks, we show improvements for large models to retain seen and zero/few shot task performance in online settings.

## 1 INTRODUCTION

The adaptation of deep neural networks have practical importance. It enables models to adapt to varying test-time distributions, attributed to shifts in time, person, environment, etc. The more difficult adaptation cases arise when there may be no clear task boundaries, when the task was not seen during training, and only few/zero samples are available to update a model. To tackle adaptation broadly, given a *base learner* optimizing its inner objective with respect to its assigned task, a *meta learner* computes the update to the base learner such that it optimizes its outer objective across a distribution of tasks (Hospedales et al., 2021). Scaling the size of models and training data have recently demonstrated comparable zero/few-shot capabilities (e.g. GPT-3 (Brown et al., 2020), Chinchilla (Hoffmann et al., 2022)). Retaining this zero/few-shot capability becomes a challenge in an online setting. Prior continual learning methods (Lange et al., 2019) aim to retain performance on both prior and subsequent tasks, but do not evaluate the retention of zero/few-shot task performance. Ilharco et al. (2022) proposed an online algorithm that fine-tunes a large vision-language model on a new task, and performs well on the previous zero/few-shot tasks and the seen fine-tuned task.

Task-specific representations within a large model can be difficult to disentangle and manipulate. Identifying and freezing subnetworks (e.g. APD (Yoon et al., 2020), WSN (Kang et al., 2022)) can help mitigate forgetting. A meta learner projects its representations onto a base parameter space. For a gradient-based meta learner, the meta learner and base parameters reside in the same parameter space. By optimizing meta parameters alike to gradient-based meta learning, we can project the task-specific representations (subnetworks) in the meta parameters to interpolatable, equidimensional base parameters (subnetworks) in one parameter space. Our proposed method, Subnetwork Projection (SNP), trains a meta learner to maximize the distance that the meta parameters can drift while returning the same base parameters. SNP++ also stores a memory buffer to access and manipulate the base parameters.

**Contributions.** Subnetwork Projection (SNP) is the first continual learner designed to retain seen and unseen zero/few-shot performance on both prior and subsequent tasks, outperforming existing baselines. By projecting subnetworks as equidimensional base parameters in the same space, SNP trains a model to sustain greater parameter drift while still retaining access to the original base parameters. With task-specific samples, SNP++ can manipulate subnetworks encoded in the meta parameters, including adding, removing, combining, or switching subnetworks.

## 2 RELATED WORK

**Discrete representations.** Identifying and freezing task-specific subnetworks can minimize forgetting on prior tasks (Yoon et al., 2020; Kang et al., 2022). A concern with this discrete representation is its mutability. Once a subnetwork is identified and frozen, it cannot be transformed or switched to a

Table 1: Measuring the cosine distance between flattened parameters fine-tuned on each dataset against each other.

|  | MSCOCO | ImageNet | CIFAR100 | STL10 | Caltech101 | StanfordCars | Flowers102 | GTSRB | Food101 | EuroSAT | FGVCAircraft |
|---|---|---|---|---|---|---|---|---|---|---|---|
| MSCOCO | 0.0000 | 0.0242 | 0.0241 | 0.0253 | 0.0242 | 0.0352 | 0.0064 | 0.0025 | 0.0298 | 0.0052 | 0.0332 |
| ImageNet | 0.0242 | 0.0000 | 0.0437 | 0.0464 | 0.0456 | 0.0572 | 0.0298 | 0.0264 | 0.0510 | 0.0289 | 0.0555 |
| CIFAR100 | 0.0241 | 0.0437 | 0.0000 | 0.0471 | 0.0452 | 0.0577 | 0.0300 | 0.0262 | 0.0508 | 0.0288 | 0.0558 |
| STL10 | 0.0253 | 0.0464 | 0.0471 | 0.0000 | 0.0470 | 0.0588 | 0.0311 | 0.0276 | 0.0536 | 0.0301 | 0.0570 |
| Caltech101 | 0.0242 | 0.0456 | 0.0452 | 0.0470 | 0.0000 | 0.0574 | 0.0298 | 0.0263 | 0.0519 | 0.0290 | 0.0556 |
| StanfordCars | 0.0352 | 0.0572 | 0.0577 | 0.0588 | 0.0574 | 0.0000 | 0.0405 | 0.0372 | 0.0622 | 0.0396 | 0.0645 |
| Flowers102 | 0.0064 | 0.0298 | 0.0300 | 0.0311 | 0.0298 | 0.0405 | 0.0000 | 0.0087 | 0.0352 | 0.0113 | 0.0386 |
| GTSRB | 0.0025 | 0.0264 | 0.0262 | 0.0276 | 0.0263 | 0.0372 | 0.0087 | 0.0000 | 0.0318 | 0.0075 | 0.0352 |
| Food101 | 0.0298 | 0.0510 | 0.0508 | 0.0536 | 0.0519 | 0.0622 | 0.0352 | 0.0318 | 0.0000 | 0.0340 | 0.0602 |
| EuroSAT | 0.0052 | 0.0289 | 0.0288 | 0.0301 | 0.0290 | 0.0396 | 0.0113 | 0.0075 | 0.0340 | 0.0000 | 0.0379 |
| FGVCAircraft | 0.0332 | 0.0555 | 0.0558 | 0.0570 | 0.0556 | 0.0645 | 0.0386 | 0.0352 | 0.0602 | 0.0379 | 0.0000 |

Table 2: Measuring the (Zero-shot Top-5 / Few-shot Top-1) accuracy between parameters fine-tuned on each dataset against each other. We baseline against the pretrained initialization trained on WIT, the finetuned model on MSCOCO (which was then used as the starting point for each subsequently-finetuned model).

| | | Evaluation dataset | | | | | | | | | | |
|---|---|---|---|---|---|---|---|---|---|---|---|---|
| | | MSCOCO (N=91) | ImageNet (N=1000) | CIFAR100 (N=100) | STL10 (N=10) | Caltech101 (N=102) | StanfordCars (N=196) | Flowers102 (N=102) | GTSRB (N=43) | Food101 (N=101) | EuroSAT (N=10) | FGVCAircraft (N=100) |
| | Pretrained init | 23.1 / 47.0 | 83.5 / 56.9 | 69.7 / 38.2 | 99.7 / 94.0 | 85.3 / 60.7 | 85.6 / 51.2 | 84.0 / 90.1 | 56.0 / 30.6 | 86.7 / 74.6 | 76.0 / 69.0 | 44.6 / 32.2 |
| | MSCOCO | 93.7 / 91.2 | 8.6 / 10.9 | 11.3 / 10.8 | 95.3 / 55.2 | 8.1 / 43.4 | 2.7 / 7.2 | 7.0 / 55.1 | 17.2 / 16.6 | 16.4 / 12.3 | 54.3 / 52.4 | 5.4 / 7.8 |
| | ImageNet | 9.8 / 20.8 | 91.2 / 89.9 | 14.0 / 11.3 | 95.5 / 59.4 | 7.8 / 9.7 | 3.1 / 7.4 | 7.1 / 57.8 | 14.9 / 12.0 | 14.4 / 10.9 | 57.4 / 56.4 | 4.9 / 8.0 |
| | CIFAR100 | 4.9 / 10.1 | 8.2 / 6.8 | 93.0 / 91.5 | 77.3 / 40.5 | 5.8 / 14.3 | 3.1 / 6.6 | 6.1 / 38.2 | 12.1 / 11.9 | 6.4 / 4.6 | 48.5 / 63.7 | 5.5 / 6.1 |
| Model trained on | STL10 | 6.0 / 3.9 | 2.6 / 3.2 | 7.1 / 7.4 | 99.7 / 91.6 | 3.8 / 4.8 | 2.9 / 5.3 | 4.9 / 27.2 | 10.8 / 11.3 | 6.1 / 4.2 | 50.5 / 51.1 | 5.0 / 5.7 |
| | Caltech101 | 7.2 / 8.2 | 1.8 / 4.8 | 8.3 / 9.2 | 86.8 / 42.9 | 98.5 / 98.4 | 2.9 / 7.4 | 5.4 / 43.7 | 12.5 / 13.8 | 5.6 / 6.3 | 53.9 / 54.0 | 4.6 / 3.0 |
| | StanfordCars | 1.7 / 8.7 | 1.9 / 5.1 | 6.5 / 7.5 | 68.0 / 41.1 | 3.1 / 5.0 | 94.0 / 95.7 | 6.3 / 47.0 | 16.2 / 18.5 | 6.7 / 5.4 | 53.9 / 52.2 | 4.8 / 12.6 |
| | Flowers102 | 1.4 / 6.3 | 3.7 / 7.7 | 5.9 / 9.4 | 88.1 / 43.2 | 7.7 / 16.1 | 2.7 / 5.6 | 99.5 / 97.3 | 9.5 / 15.2 | 9.0 / 8.9 | 52.7 / 53.3 | 5.8 / 8.4 |
| | GTSRB | 23.4 / 5.1 | 1.8 / 6.0 | 14.0 / 11.0 | 72.9 / 40.7 | 7.0 / 11.1 | 2.5 / 5.3 | 7.0 / 43.6 | 99.5 / 97.8 | 6.5 / 6.3 | 44.4 / 53.2 | 4.8 / 8.8 |
| | Food101 | 27.2 / 5.2 | 1.4 / 6.0 | 6.8 / 10.9 | 69.1 / 37.4 | 5.8 / 5.5 | 2.5 / 5.8 | 5.0 / 52.4 | 7.9 / 12.6 | 93.2 / 97.4 | 48.9 / 58.8 | 5.7 / 6.6 |
| | EuroSAT | 35.0 / 3.2 | 6.5 / 1.9 | 6.8 / 5.8 | 67.2 / 28.6 | 9.7 / 16.8 | 2.7 / 6.9 | 6.3 / 8.5 | 11.7 / 9.9 | 4.3 / 2.7 | 97.3 / 95.8 | 5.3 / 2.4 |
| | FGVCAircraft | 4.2 / 4.0 | 1.7 / 5.0 | 7.6 / 7.1 | 61.1 / 29.0 | 4.1 / 6.6 | 2.3 / 3.2 | 6.2 / 44.7 | 12.9 / 12.0 | 8.5 / 6.2 | 55.2 / 56.7 | 89.5 / 99.3 |

different subnetwork if its source data is unavailable. This would be needed if most of the network is frozen and no leftover nodes are available for a new task. Task order affects the subnetwork arrangement, and changes in unfrozen subnetworks values may render the frozen subnetwork inaccurate. Network capacity would also be fixed; once all the nodes are frozen, a subnetwork would need to be removed in order for a new task to be learnt. Interpolation between subnetworks may not viable due to different shapes. Subnetworks could also be constructed as stitchable layers of regular shapes (e.g. linear layer), such as in model stitching (Csiszárik et al., 2021; Bansal et al., 2021) or feature adapters (Gao et al., 2021; Chen et al., 2022). These layers would need to be compatible and conditioned on the previous layers. Networks can also be modularly generated from architectural components (Andreas et al., 2016a;b).

**Continuous representations.** Instead of manipulating discrete, composable modules in neural networks, the weights/parameters of the network can be modified. Combining representations is a common technique, aiding in leveraging transferable properties between tasks as well as conserving capacity. Regularization-based continual learning strategies, such as EWC (Kirkpatrick et al., 2017) and SI (Zenke et al., 2017), use regularization terms to update parameters towards the new task while retaining pertinent representations of the prior task. Model merging and averaging has also been used to improve generalizability, robustness, and adaptatability in online learning settings (Singh & Jaggi, 2019; Matena & Raffel, 2021; Wortsman et al., 2022; Ilharco et al., 2022). Other than interpolatability, residing in a continuous space enables these representations to be dynamically-generated. Meta learners (e.g. MAML (Finn et al., 2017b), hypernetworks (Ha et al., 2016)) query a few samples from the task to compute the updated base parameters. Large models can support similar few-shot capabilities with parameter-efficient fine-tuning, such as prompt tuning (Sanh et al., 2022; Wei et al., 2022a;b; Zhou et al., 2022).

Table 3: Empirical equivalence between CLIP and MAML(CLIP). Both networks have the same number of parameters and architecture, but vary by training optimization procedure. They can both be trained to attain comparable (Zero-shot Top-5 / Few-shot Top-1) accuracy.

| | #params | MSCOCO (N=91) | ImageNet (N=1000) | CIFAR100 (N=100) | STL10 (N=10) | Caltech101 (N=102) | StanfordCars (N=196) | Flowers102 (N=102) | GTSRB (N=43) | Food101 (N=101) | EuroSAT (N=10) | FGVCAircraft (N=100) |
|---|---|---|---|---|---|---|---|---|---|---|---|---|
| CLIP | 102,007,137 | 93.7 / 91.2 | 8.6 / 10.9 | 11.3 / 10.8 | 95.3 / 55.2 | 8.1 / 43.4 | 2.7 / 7.2 | 7.0 / 55.1 | 17.2 / 16.6 | 16.4 / 12.3 | 54.3 / 52.4 | 5.4 / 7.8 |
| MAML(CLIP) | 102,007,137 | 94.6 / 95.8 | 9.8 / 12.1 | 13.3 / 14.6 | 89.1 / 50.4 | 10.8 / 37.8 | 2.3 / 6.4 | 7.4 / 68.5 | 19.2 / 23.3 | 11.4 / 16.6 | 52.8 / 57.6 | 4.5 / 7.3 |

## 3 GROUNDING SUBNETWORK PROJECTION IN META LEARNERS

**Problem Setup.** From a task set $\mathbb{T} = \{\mathcal{T}_t\}^{t \in T}$, a base learner receives $T$ tasks sequentially. $\mathcal{T}_t = \{x_t, y_t\}$ denotes the dataset of the $t$-th task. In the online/continual learning setting, given loss function $\mathcal{L}$, a base learner $f(\theta_{\text{base}}; x)$ optimizes its parameters $\theta_{\text{base}}$ such that it can perform well on the $t$-th task while minimizing performance drop on the previous $(t-1)$ tasks. We further add the requirement of retaining the zero/few-shot performance of unseen tasks $\mathbb{V}_t = \{\mathcal{V}_{t,v}\}^{v \in V}$ of the $t$-th seen task. Hence, the objective is: $\theta_{\text{base}}^* := \arg\min_{\theta_{\text{base}}} \sum_{t=1}^{T} [\mathcal{L}(f(\theta_{\text{base}}; x_t), y_t) + \sum_{v=1}^{V} \mathcal{L}(f(\theta_{\text{base}}; x_v), y_v)]$. We measure drift between two parameters with the distance function $\texttt{dist}(\theta_0, \theta_1)$. The experimental setup is described in Section 6.1.

---

**Algorithm 1** `base_params`

---

1: **procedure** `base_params`$(\theta, \mathbb{T} = \{\mathcal{T}_t\}^{t \in T}, K, lr_{\text{base}})$
2:     **for** $\mathcal{T}_t$ in $\mathbb{T}$ **do**
3:         **for** $X_t^K, Y_t^K$ in $\mathcal{T}_t$ **do**
4:             $\theta_{\text{base},t} = \theta - lr_{\text{base}} \frac{\partial \mathcal{L}(\theta; X_t^K, Y_t^K)}{\partial \theta}$
5:     **return** $\{\theta_{\text{base},t}\}^{t \in T}$

---

### 3.1 TASK GROUPINGS

The pre-trained model was trained on WebImageText (WIT) (Radford et al., 2021), a dataset specially gathered with 400M (image, text) pairs, in contrast to 100K in MSCOCO (Lin et al., 2014). From the pre-trained initialization, we train on the first task of MSCOCO for 50 epochs, until the accuracy is high on MSCOCO and low for all the unseen datasets that the pre-trained model performed well on. We reuse 10 datasets used in CLIP's evaluation including ImageNet (Deng et al., 2009), CIFAR100 (Krizhevsky, 2009), STL-10 (Coates et al., 2011), Caltech-101 (Fei-Fei et al., 2004), Stanford Cars (Krause et al., 2013), Oxford Flowers 102 (Nilsback & Zisserman, 2008), GTSRB (Stallkamp et al., 2012), Food-101 (Bossard et al., 2014), EuroSAT (Helber et al., 2017), and FGVC Aircraft (Maji et al., 2013). While MSCOCO contains (image, caption) pairs, the other datasets contain (image, label) pairs. Hence, with the prompt templates provided by Radford et al. (2021) for each dataset (e.g. "a photo of a CLASS", we can convert labels to captions.

While the pre-trained initialization performs well across the 11 datasets, the MSCOCO-finetuned model loses many transferable representations from WIT, such that the average zero/few-shot accuracy is low. MSCOCO is on a much smaller scale than WIT, in terms of number of images (100K vs 400M), range of classes (e.g. ImageNet labels such as stingray, tarantula, mousetrap are not found in MSCOCO), and less diverse images (e.g. MSCOCO contains natural and day-to-day scenes, while WIT contains natural scenes, sketches, blurry images, low-res images, texts, websites). The lifelong goal is to gradually increase the average zero/few-shot accuracy across all tasks. Given the range of datasets, we evaluate zero/few-shot transferability between them, such that learning one dataset will yield high zero/few-shot performance on another dataset. First, we fine-tuned CLIP on each dataset from a MSCOCO-finetuned initialization.

In Table 1, we measured the cosine distance between each pair of models fine-tuned on two different datasets. This infers the spatial distance in the parameter space, which parameters are closer to each other, and which parameters require further gradient updates from the initialization. We are able to identify 3 sets of datasets, grouped by distance: (i) $\leq 0.1$, (ii) $0.1 - 0.3$, (iii) $\geq 0.3$. In Table 2, we evaluate the functional distance of each fine-tuned model by computing the zero/few-shot performance

Table 4: After drifting the meta parameters to each task's base parameters, the newly-computed base parameters for each task has minimal drift with respect to the original base parameters.

| Computed base parameters | Distance w.r.t. base parameters of task=1 as meta parameters | Distance w.r.t. base parameters of task=2 as meta parameters | Distance w.r.t. base parameters of task=3 as meta parameters | Distance w.r.t. base parameters of task=4 as meta parameters | Distance w.r.t. base parameters of task=5 as meta parameters |
|---|---|---|---|---|---|
| Task 1 | 0.0937 | 0.1327 | 0.1202 | 0.1598 | 0.1120 |
| Task 2 | 0.1426 | 0.0263 | 0.1370 | 0.1810 | 0.1319 |
| Task 3 | 0.1667 | 0.1368 | 0.1008 | 0.1798 | 0.1385 |
| Task 4 | 0.1439 | 0.1600 | 0.1341 | 0.0181 | 0.1511 |
| Task 5 | 0.1338 | 0.1336 | 0.1238 | 0.1164 | 0.0942 |

per model on each dataset. Based on the relational analysis between seen and unseen tasks, given the distance between models fine-tuned on each dataset (relative cosine distance in the parameter space), and given the zero/few-shot performance of each fine-tuned model, some clear task groupings can be identified. The order for the seen (unseen) tasks is: MSCOCO (FGVCAircraft, EuroSAT, Food101) → ImageNet (STL10, StanfordCars) → Caltech101 (CIFAR100, GTSRB, Flowers102).

## 3.2 DISENTANGLING MODEL REPRESENTATIONS AS BASE PARAMETERS

Meta learners are trained to develop zero/few-shot capabilities. Given meta parameters, some meta learners map input distributions to base parameters. The base parameters are task-specific representations, and dynamically generated with respect to the task. In the case of gradient-based meta learners, the meta parameters and base parameters reside in the same parameter space. A gradient is computed with respect to the new task's input distribution and applied to the meta parameters, and this returns the base parameters. As a result, we can project the task-specific representations and subnetworks within the meta parameters to the base parameter space, and use a gradient-based meta learner to retain the same model architecture and output space as training a model without a meta learning training procedure. We train CLIP with MAML's (Finn et al., 2017a) training procedure for 10,000 epochs, meta learning rate and base learning rate of 0.0005. To retain the same scale of model and data, we use the same CLIP architecture and capacity, and retain the same dataset size by training MAML(CLIP) on Split-MSCOCO (Chiaro et al., 2020) (which organizes labelled MSCOCO into 5 tasks: transport, animals, sports, food, interior). We train MAML(CLIP) until it attains similar zero/few-shot performance as CLIP (Table 3).

Table 5: Measuring the Euclidean distance between the original meta parameters of MAML(CLIP) and each task's base parameters is indicative of the subspace radius.

| Computed base parameters | Distance w.r.t. meta parameters |
|---|---|
| Task 1 | 0.388 |
| Task 2 | 0.413 |
| Task 3 | 0.340 |
| Task 4 | 0.420 |
| Task 5 | 0.310 |

## 3.3 DRIFT IN A META PARAMETER'S SUBSPACE

When updating the meta learning parameters $\theta$ on a new task, $\theta$ may drift by some distance to $\theta'$. Given base parameters $\{\theta_{\text{base},t}\}^{t \in T}$ and $\theta$ reside in the same parameter space, we evaluate how far $\theta$ can drift while returning the same (or within bounded error $\varepsilon$) base parameters $\text{dist}(\theta_{\text{base},t}, \theta'_{\text{base},t}) \leq \varepsilon$.

From Table 5, we first measure the Euclidean distance between the original meta parameters and their MAML-computed base parameters. This informs the approximate radius of the parameter subspace. In Table 4, we test whether drifting the meta parameter to each task's base parameter $\theta = \theta_{\text{base},t}$ will still be able to compute the same task base parameters. Relative to the subspace radius, we find that the base parameters can indeed be re-located if the meta parameter is drifted to the end-points of the subspace. Given that the base parameters can be located if the drift is within the bounds of the subspace, we next evaluate whether the base parameters can be located if the drift exceeds the bounds of the subspace. In Table 6, we evaluate $S = 1000$ random parameters, and interpolate between this random parameter $\theta_{\text{rand},s}$ and the meta parameter $\theta$ to return an interpolated meta parameter $\theta_{\text{int}} = (1 - r)\theta + r\theta_{\text{rand},s}$ with interpolation coefficient $r$. We find that for varying interpolation coefficients (and thus varying Euclidean distances), once the Euclidean distance increases substantially, then the computed base parameters drift in a similar fashion from the original base parameters. As a result, we are interested in maximizing the radius of the parameter subspace in which the meta parameter can drift, while returning the same base parameters (within bounded error).

---

**Algorithm 2** `adaptive_beta`

---

1: **procedure** `adaptive_beta`$(\beta_{\text{meta}}, \text{dist}_{\text{meta},v}, \varepsilon, \{\text{dist}_{\text{meta},s,r}\}^{s,r \in S,I}, \mathbb{T}, K, lr_{\text{base}})$
2:      **if** $\varepsilon = $ None **then**
3:          $\{\theta_{\text{base},t}\} \leftarrow$ `base_params`$(\theta, \mathbb{T}, K, lr_{\text{base}})$
4:          dist_list $= \{\}$
5:          **for** $\theta^{'}$ in $\{\theta_{\text{base},t}\}$ **do**
6:              $\{\theta^{'}_{\text{base},t}\} \leftarrow$ `base_params`$(\theta^{'}, \mathbb{T}, K, lr_{\text{base}})$
7:              dist_list $\leftarrow \frac{1}{T}\Sigma_t^T$ `dist`$(\theta^{'}_{\text{base},t}, \theta_{\text{base},t})$
8:          $\varepsilon = \max(\text{dist\_list})$
9:      $\text{dist}_{\text{meta}} := \arg\max_{s,r}[\text{dist}_{\text{meta},s,r} | \text{dist}_{\text{base},s,r} \leq \varepsilon]$
10:      $\beta_{\text{meta}} = \max(\beta_{\text{meta}}, \beta_{\text{meta}} \times \frac{\text{dist}_{\text{meta}}}{\text{dist}_{\text{meta}} - \text{dist}_{\text{meta},v}})$
11:      **return** $\beta_{\text{meta}}$

---

**Algorithm 3** `train_space`

---

1: **procedure** `train_space`$(\mathbb{T} = \{\mathcal{T}_t\}^{t \in T}, K = 5,$    epochs $= 10,000, lr_{\text{base}} = 0.0005, lr_{\text{meta}} = 0.0005, \beta_{\text{meta}} = 0.5, \beta_{\text{base}} = \{\beta_{\text{base},t} = 0.5\}^{t \in T}, S = 1,000, I = [0.0001, 0.001, 0.01, 0.1], \mathcal{M} = $ False or $\{\})$
2:      $\theta \leftarrow \theta_{\text{init}}$
3:      **if** $\mathcal{M} \neq$ False **then**                                      ▷ optional: store memory
4:          $\mathcal{M} \leftarrow \{X_t^K, Y_t^K\}^{t \in T}$
5:      **for** epoch in epochs **do**
6:          $\{\theta_{\text{base},t}\} \leftarrow$ `base_params`$(\theta, \mathbb{T}, K, lr_{\text{base}})$
7:          **for** $\mathcal{T}_t$ in $\mathbb{T}$ **do**
8:              **for** $X_t, Y_t$ in $\mathcal{T}_t$ **do**
9:                  $\mathcal{L}_{\mathcal{T}_t} = \mathcal{L}(\theta - lr_{\text{base}}\frac{\partial\mathcal{L}(\theta;X_t,Y_t)}{\partial\theta}; X_t, Y_t)$
10:          **for** $s$ in $S$ **do**
11:              $\theta_{\text{rand}} \leftarrow \theta_{\text{init}}$
12:              **for** $r$ in $I$ **do**
13:                  $\theta_{\text{int}} = (1-r)\theta + r\theta_{\text{rand}}$
14:                  $\text{dist}_{\text{meta},s,r} = $ `dist`$(\theta, \theta_{\text{int}})$
15:                  $\{\theta^{\text{int}}_{\text{base},t}\} \leftarrow$ `base_params`$(\theta_{\text{int}}, \mathbb{T}, K, lr_{\text{base}})$
16:                  $\text{dist}_{\text{base},s,r} = \Sigma_t^T$ `dist`$(\theta_{\text{base},t}, \theta^{\text{int}}_{\text{base},t})$
17:          $\mathcal{L}_{\text{meta}} = \Sigma_s^S \Sigma_r^I \text{dist}_{\text{meta},s,r}$
18:          $\mathcal{L}_{\text{base}} = \Sigma_s^S \Sigma_r^I \text{dist}_{\text{base},s,r}$
19:          $\theta := \theta - lr_{\text{meta}}\Sigma_t^T \frac{\partial\mathcal{L}_{\mathcal{T}_t}}{\partial\theta} - \beta_{\text{meta}}\frac{\partial\mathcal{L}_{\text{meta}}}{\partial\theta} - \beta_{\text{base}}\frac{\partial\mathcal{L}_{\text{base}}}{\partial\theta}$
20:      **return** $\theta, \mathcal{M}$

---

## 4   SUBNETWORK PROJECTION (SNP): EXPAND PROJECTED SUBSPACE TO SUPPORT DRIFT

Given a model architecture, we can alter the training procedure to one of gradient-based meta learning and project the subnetworks onto a base learner's parameter space. In the standard implementation, we assume no memory $\mathcal{M} = $ False. We cannot access subnetworks, but we can regulate the training of the meta parameters such that we can maximize the radius of the parameter subspace in which the meta parameter can drift, while returning the same base parameters within bounded error (Algorithms 3-4). Referring to Algorithm 3, per epoch, after computing the support set loss w.r.t. computed base parameters, we compute a set of distance regularization terms. Our selected distance function `dist` is cosine distance. We sample interpolated meta parameters at varying distances from the current epoch's meta parameters, and compute the cumulative drift in meta

Table 6: Interpolating the meta parameters against randomly-sampled distant parameters, we find that the closer the interpolated meta parameters, the closer the Euclidean distance, and within certain limits of drift, the base parameters would drift minimally.

| Interpolation coefficient | Meta parameter drift | Base parameter drift (avg across tasks) |
|---|---|---|
| 0.001 | 0.327 | 0.0839 |
| 0.1 | 33 | 33 |
| 1 | 330 | 330 |

---

**Algorithm 4 expand_space**

---

1: **procedure** expand_space($\theta$, $\mathbb{V} = \{\mathcal{V}_v\}^{v \in V}$, $K = 5$, epochs $= 500$, $lr_{\text{base}} = 0.0005$, $lr_{\text{meta}} = 0.0005$
$\quad \beta_{\text{meta}} = 0.5$, $\beta_{\text{base}} = \{\beta_{\text{base},v} = 0.5\}^{v \in V}$, $\beta_{\text{int}} = \{\beta_{\text{int},v} = 1.0\}^{v \in V}$, $\mathcal{M} = \texttt{False}$ or $\{X_t^K, Y_t^K\}^{t \in T}$,
$\quad \varepsilon = 0.001$ or None)
2: $\quad$ **if** $\mathcal{M} \neq \texttt{False}$ **then** $\hfill \triangleright$ optional: access subnetworks
3: $\quad\quad$ **for** $X_t^K, Y_t^K$ in $\mathcal{M}$ **do**
4: $\quad\quad\quad$ $\theta_{\text{base},t}^* = \theta - lr_{\text{base}} \frac{\partial \mathcal{L}(\theta; X_t^K, Y_t^K)}{\partial \theta}$
5: $\quad$ **for** $\mathcal{V}_v$ in $\mathbb{V}$ **do**
6: $\quad\quad$ **for** epoch in epochs **do**
7: $\quad\quad\quad$ $\{\theta_{\text{base},v}\} \leftarrow \texttt{base\_params}(\theta, \mathbb{V}, K, lr_{\text{base}})$
8: $\quad\quad\quad$ **if** $\mathcal{M} \neq \texttt{False}$ **then**
9: $\quad\quad\quad\quad$ **if** $\beta_{\text{int},v} > 0$ **then**
10: $\quad\quad\quad\quad\quad$ **for** $\theta_{\text{base},v}$ in $\{\theta_{\text{base},v}\}$ **do**
11: $\quad\quad\quad\quad\quad\quad$ $g := \arg\min_{g \in T} \texttt{dist}(\theta_{\text{base},v}, \theta_{\text{base},g}^*)$
12: $\quad\quad\quad$ **for** $X_v, Y_v$ in $\mathcal{V}_v$ **do**
13: $\quad\quad\quad\quad$ $\mathcal{L}_{\mathcal{V}_v} = \mathcal{L}(\theta - lr_{\text{base}} \frac{\partial \mathcal{L}(\theta; X_v, Y_v)}{\partial \theta}; X_v, Y_v)$
14: $\quad\quad\quad$ $\texttt{dist}_{\text{meta},v} = \texttt{dist}(\theta, \theta - lr_{\text{meta}} \Sigma_v^V \frac{\partial \mathcal{L}_{\mathcal{V}_v}}{\partial \theta})$
15: $\quad\quad\quad$ **if** $\mathcal{M} \neq \texttt{False}$ **then**
16: $\quad\quad\quad\quad$ **for** $X_t^K, Y_t^K$ in $\mathcal{M}$ **do**
17: $\quad\quad\quad\quad\quad$ **if** $\beta_{\text{base},t} > 0$ **then**
18: $\quad\quad\quad\quad\quad\quad$ $\texttt{dist}_{\text{base},t} = \texttt{dist}(\theta_{\text{base},t}^*,$
$\quad\quad\quad\quad\quad\quad\quad\quad \theta - lr_{\text{base}} \frac{\partial \mathcal{L}(\theta; X_t^K, Y_t^K)}{\partial \theta})$
19: $\quad\quad\quad$ **if** $\mathcal{M} \neq \texttt{False}$ **then** $\hfill \triangleright$ optional: interp./remove
20: $\quad\quad\quad\quad$ **if** $\beta_{\text{int},v} > 0$ **then**
21: $\quad\quad\quad\quad\quad$ $X_g^K, Y_g^K \leftarrow \mathcal{M}$
22: $\quad\quad\quad\quad\quad$ $X_v^K, Y_v^K \leftarrow \mathcal{V}_v$
23: $\quad\quad\quad\quad\quad$ $\texttt{dist}_{\text{int}} = \texttt{dist}(\theta - lr_{\text{base}} \frac{\partial \mathcal{L}(\theta; X_v^K, Y_v^K)}{\partial \theta},$
$\quad\quad\quad\quad\quad\quad\quad\quad \theta - lr_{\text{base}} \frac{\partial \mathcal{L}(\theta; X_g^K, Y_g^K)}{\partial \theta})$
24: $\quad\quad\quad$ $\beta_{\text{meta}} \leftarrow \texttt{adaptive\_beta}(\beta_{\text{meta}}, \texttt{dist}_{\text{meta},v}, \varepsilon)$
25: $\quad\quad\quad$ $\mathcal{L}_{\text{meta}} = \texttt{dist}_{\text{meta},v}$; $\mathcal{L}_{\text{int}} = \texttt{dist}_{\text{int}}$
26: $\quad\quad\quad$ $\mathcal{L}_{\text{base}} = \Sigma_t^T \texttt{dist}_{\text{base},t}$
27: $\quad\quad\quad$ $\theta := \theta - lr_{\text{meta}} \Sigma_v^V \frac{\partial \mathcal{L}_{\mathcal{V}_v}}{\partial \theta}$
$\quad\quad\quad\quad -\beta_{\text{meta}} \frac{\partial \mathcal{L}_{\text{meta}}}{\partial \theta} - \beta_{\text{base}} \frac{\partial \mathcal{L}_{\text{base}}}{\partial \theta} - \beta_{\text{int},g} \frac{\partial \mathcal{L}_{\text{int}}}{\partial \theta}$
28: $\quad\quad$ **if** $\mathcal{M} \neq \texttt{False}$ **then**
29: $\quad\quad\quad$ $\mathcal{M} \leftarrow \{X_v^K, Y_v^K\}^{v \in V}$
30: $\quad$ **return** $\theta, \mathcal{M}$

---

parameters and base parameters. With these loss terms, we update the meta parameters. In an online setting (Algorithm 4), we perform distance regularization on the meta parameters (but not the base parameters, as $\mathcal{M} = \texttt{False}$). Given knowledge of the subspace radius from the training procedure, while we measure the drift of the meta parameters, we are informed on when the base parameters error will increase (e.g. exceeding the radius). As such, we make use of an adaptive regularization coefficient procedure (Algorithm 2): when meta parameters are closer to the end of the supported radius, the distance regularization coefficient will increase accordingly.

## 5 SNP++: MEMORY-BASED SUBNETWORK ACCESS AND MANIPULATION

To query a subnetwork, we need task-specific data, in-line with prior subnetwork literature. Unlike replay-based methods, we do not store extensive replay buffers; instead, the memory buffer is one instance of a N-way-K-shot support set for computing base parameters. The use of this task-specific support introduces various interesting properties for manipulating the model. As the support set varies, we can map each input distribution to a unique subnetwork. As such, we have a continuous space of subnetworks. In the standard case, we intend to add new subnetworks. First we use the previous training procedure to maximize the subspace radius. For each new task, we can fine-tune our meta parameters w.r.t. the new dataset, while using the memory buffer to track and regularize the drift

of each individual base parameter. Unlike SNP, we regularize both the drift in the meta parameters as well as each base parameter.

Table 7: Moving from pre-trained initialization, to Task 1-3, we present the (Zero-shot Top-5 / Few-shot Top-1) accuracy across each task and baseline method. We also measure backward transfer (BWT) (Lopez-Paz & Ranzato, 2017), which is the influence that learning a new task has on the performance on a previous task. Positive backward transfer occurs when learning a new task increases the performance on a preceding task. Negative backward transfer occurs when learning about a new task decreases the performance on a preceding task. Rather than computing a general BWT score, for those tasks that are greater than or equal to its compared value, we compute the average positive BWT. For those tasks that are less than its compared value, we compute the average negative BWT. This helps us measure positive transfer as well as drawdown. For Task 3, we evaluate each method against their method's task 2 performance; otherwise, each task's method is evaluated against the previous task's fine-tuning performance.

| | Task 1: MSCOCO | Task 2: ImageNet | | | | | | | | |
| --- | --- | --- | --- | --- | --- | --- | --- | --- | --- | --- |
| | Fine-tuning | Fine-tuning | EWC | BatchEnsemble | GPM | CLIP-Adapter | PAINT | SNP | SNP++ (Add.) | SNP++ (Inter.) |
| MSCOCO (N=91) | 93.7 / 91.2 | 51.2 / 54.8 | 78.3 / 84.7 | 87.4 / 86.6 | 77.5 / 86.8 | 80.5 / 81.6 | 86.0 / 89.9 | 88.2 / 89.3 | 90.8 / 86.0 | 86.3 / 92.4 |
| ImageNet (N=1000) | 8.6 / 10.9 | 83.2 / 31.3 | 71.4 / 28.6 | 74.6 / 29.0 | 72.7 / 28.6 | 75.9 / 27.6 | 81.0 / 31.3 | 78.6 / 36.4 | 80.9 / 35.1 | 76.9 / 37.7 |
| CIFAR100 (N=100) | 11.3 / 10.8 | 15.8 / 9.2 | 11.5 / 10.2 | 11.4 / 10.4 | 11.4 / 10.1 | 11.7 / 10.4 | 15.0 / 10.7 | 14.7 / 11.3 | 15.1 / 10.9 | 14.4 / 11.7 |
| STL10 (N=10) | 95.3 / 55.2 | 97.2 / 61.7 | 95.4 / 54.9 | 91.3 / 56.2 | 88.0 / 57.6 | 96.3 / 56.6 | 96.1 / 55.9 | 94.5 / 52.4 | 97.3 / 50.5 | 92.5 / 54.2 |
| Caltech101 (N=102) | 8.1 / 43.4 | 8.2 / 50.5 | 8.3 / 46.8 | 4.9 / 44.9 | 7.7 / 46.2 | 9.1 / 45.0 | 8.2 / 45.1 | 9.4 / 47.9 | 9.7 / 46.1 | 9.2 / 49.6 |
| StanfordCars (N=196) | 2.7 / 7.2 | 3.1 / 7.3 | 3.4 / 7.3 | 2.2 / 7.3 | 3.6 / 7.2 | 3.5 / 7.1 | 3.9 / 7.3 | 3.7 / 7.6 | 3.8 / 7.3 | 3.6 / 7.9 |
| Flowers102 (N=102) | 7.0 / 55.1 | 6.7 / 60.8 | 6.5 / 56.3 | 6.8 / 59.0 | 6.7 / 54.8 | 7.1 / 55.2 | 6.9 / 59.4 | 7.4 / 63.7 | 7.6 / 61.4 | 7.2 / 65.9 |
| GTSRB (N=43) | 17.2 / 16.6 | 15.6 / 17.2 | 16.7 / 16.7 | 16.2 / 17.1 | 16.7 / 17.2 | 16.9 / 17.2 | 15.1 / 17.2 | 16.1 / 16.9 | 16.6 / 16.3 | 15.8 / 17.5 |
| Food101 (N=101) | 16.4 / 12.7 | 13.7 / 8.5 | 14.1 / 10.1 | 15.9 / 11.4 | 15.5 / 9.8 | 15.0 / 11.1 | 16.3 / 10.5 | 15.8 / 11.2 | 16.3 / 10.8 | 15.5 / 11.6 |
| EuroSAT (N=10) | 54.3 / 52.4 | 51.4 / 60.1 | 53.2 / 54.8 | 51.8 / 57.3 | 53.5 / 56.9 | 48.1 / 52.8 | 51.1 / 60.5 | 55.7 / 59.1 | 57.4 / 56.9 | 54.5 / 61.1 |
| FGVCAircraft (N=100) | 5.4 / 7.8 | 5.6 / 10.6 | 5.5 / 8.4 | 5.6 / 9.3 | 5.3 / 7.4 | 5.5 / 7.7 | 6.0 / 10.4 | 6.3 / 8.8 | 6.5 / 8.5 | 6.2 / 9.1 |
| Avg | 29.1 / 33.0 | 32.0 / 33.8 | 33.1 / 34.4 | 33.8 / 35.3 | 32.6 / 34.8 | 33.6 / 33.9 | 35.1 / 36.2 | 35.5 / 36.8 | 36.5 / 35.4 | 34.7 / 38.1 |
| Pos BWT | 0.0 / 0.0 | 13.6 / 6.4 | 10.7 / 3.6 | 16.7 / 3.9 | 21.7 / 5.6 | 10.1 / 3.5 | 13.1 / 4.8 | 11.2 / 5.9 | 10.7 / 5.5 | 10.7 / 6.4 |
| Neg BWT | 0.0 / 0.0 | -10.0 / -13.9 | -4.0 / -2.4 | -2.1 / -2.0 | -3.3 / -1.4 | -5.3 / -2.3 | -2.6 / -1.1 | -2.0 / -1.9 | -1.2 / -2.9 | -3.1 / -0.8 |

| | Task 3: Caltech101 | | | | | | | | | |
| --- | --- | --- | --- | --- | --- | --- | --- | --- | --- | --- |
| | Fine-tuning | Joint Training | EWC | BatchEnsemble | GPM | CLIP-Adapter | SNP | PAINT | SNP++ (Add.) | SNP++ (Inter.) |
| MSCOCO (N=91) | 29.1 / 37.8 | 37.3 / 39.4 | 68.6 / 69.4 | 84.2 / 86.7 | 71.7 / 73.6 | 76.7 / 79.2 | 71.1 / 71.9 | 78.2 / 81.3 | 79.0 / 80.1 | 71.5 / 83.1 |
| ImageNet (N=1000) | 15.7 / 12.9 | 80.9 / 28.6 | 63.7 / 24.9 | 71.1 / 24.6 | 67.1 / 21.8 | 70.3 / 25.4 | 77.0 / 25.0 | 76.5 / 27.3 | 77.3 / 26.9 | 69.9 / 27.9 |
| CIFAR100 (N=100) | 6.9 / 12.7 | 12.1 / 12.0 | 10.3 / 9.8 | 10.1 / 8.8 | 7.8 / 12.2 | 10.1 / 8.6 | 14.6 / 11.0 | 13.9 / 10.0 | 12.6 / 10.4 | |
| STL10 (N=10) | 91.5 / 45.3 | 95.2 / 62.9 | 73.1 / 38.7 | 83.6 / 47.3 | 88.2 / 47.8 | 91.4 / 53.9 | 91.5 / 44.7 | 90.6 / 46.4 | 91.6 / 45.7 | 82.8 / 47.4 |
| Caltech101 (N=102) | 96.1 / 93.7 | 88.2 / 89.7 | 85.0 / 92.2 | 86.5 / 85.0 | 88.8 / 84.2 | 87.1 / 82.2 | 82.4 / 70.2 | 89.1 / 91.2 | 90.1 / 89.8 | 81.4 / 93.3 |
| StanfordCars (N=196) | 2.9 / 4.6 | 3.2 / 6.3 | 3.1 / 3.6 | 2.2 / 6.1 | 3.1 / 5.1 | 2.9 / 5.3 | 3.6 / 5.8 | 3.6 / 6.0 | 3.6 / 5.9 | 3.3 / 6.1 |
| Flowers102 (N=102) | 4.7 / 47.8 | 5.8 / 61.6 | 5.9 / 43.6 | 6.6 / 48.0 | 5.1 / 49.2 | 5.6 / 55.5 | 6.6 / 47.5 | 5.9 / 50.1 | 6.0 / 49.3 | 5.4 / 51.2 |
| GTSRB (N=43) | 10.5 / 9.1 | 14.7 / 16.9 | 11.6 / 10.6 | 15.5 / 10.6 | 11.7 / 10.7 | 12.4 / 10.6 | 14.4 / 13.8 | 13.9 / 14.2 | 14.1 / 14.0 | 12.7 / 14.5 |
| Food101 (N=101) | 8.2 / 6.3 | 11.3 / 10.0 | 6.7 / 7.0 | 13.6 / 9.5 | 11.7 / 7.0 | 14.6 / 11.1 | 15.7 / 8.4 | 15.5 / 9.8 | 15.7 / 9.7 | 14.2 / 10.0 |
| EuroSAT (N=10) | 48.8 / 46.4 | 53.9 / 62.6 | 41.9 / 56.5 | 51.5 / 52.1 | 49.8 / 48.5 | 49.5 / 56.5 | 50.8 / 48.4 | 54.4 / 55.7 | 55.0 / 54.9 | 49.7 / 57.0 |
| FGVCAircraft (N=100) | 5.4 / 6.8 | 5.4 / 9.3 | 6.0 / 6.9 | 5.3 / 7.7 | 5.4 / 7.0 | 5.6 / 7.7 | 5.8 / 8.3 | 6.1 / 8.1 | 6.2 / 8.0 | 5.6 / 8.3 |
| Avg | 29.1 / 29.4 | 37.1 / 36.3 | 34.2 / 33.0 | 39.1 / 35.4 | 38.7 / 31.5 | 37.1 / 33.4 | 38.1 / 32.0 | 40.7 / 36.4 | 41.1 / 35.8 | 37.2 / 37.2 |
| Pos BWT | 87.9 / 23.3 | 27.6 / 8.0 | 38.6 / 23.6 | 39.1 / 20.1 | 27.1 / 20.0 | 26.5 / 8.2 | 74.2 / 25.1 | 79.7 / 43.3 | 80.4 / 43.7 | 72.2 / 43.7 |
| Neg BWT | -12.0 / -10.6 | -3.3 / -4.2 | -7.3 / -7.0 | -2.2 / -4.7 | -4.0 / -6.2 | -2.9 / -2.8 | -2.7 / -7.1 | -2.3 / -4.8 | -3.0 / -3.9 | -4.5 / -5.3 |

Extending further from subnetwork addition, we also can evaluate subnetwork removal, combining (or interpolating between) subnetworks, and even switching subnetworks to alternate subnetwork modes. For subnetwork removal, we can choose not to freeze/regularize a specific task's subnetwork (e.g. setting its regularization coefficient to be less then 1.0 for partial removal, or even setting to 0.0 to ignore its regularization). It does not actively remove the subnetwork, but it also does not actively preserve it. A use case is if a particular subnetwork causes interference, or if the capacity is needed for another task. In these cases, a new task's base parameter can overwrite this base parameter.

For interpolating between subnetworks, other than adding a new subnetwork entirely, we can save capacity and allow one task's base parameters to be used in multiple tasks. We can first evaluate which

Table 8: Variations in hyperparameters and subnetwork manipulation strategies with SNP(++).

| | Avg accuracy after Task 2 | Avg accuracy after Task 3 |
| --- | --- | --- |
| Rehearsal-free ablations | | |
| SNP ($\beta_{meta}$=0.1 $\beta_{base}$=1.0) | 33.9 / 35.8 | 40.2 / 34.6 |
| SNP ($\beta_{meta}$=0.5 $\beta_{base}$=0.5) | 35.5 / 36.8 | 40.7 / 36.4 |
| SNP ($\beta_{meta}$=1.0 $\beta_{base}$=0.1) | 36.5 / 38.1 | 39.4 / 36.0 |
| Addition/Removal of subnetworks | | |
| SNP++ (Addition; $\beta_{meta}$=0.5 $\beta_{base}$=[1.0, 1.0, 1.0, 1.0, 1.0]) | 36.5 / 35.4 | 41.1 / 35.8 |
| SNP++ (Partial Removal; $\beta_{base}$=[1.0, 1.0, 1.0, 1.0, 0.5]) | 35.9 / 34.7 | 36.5 / 37.7 |
| SNP++ (Removal x1; $\beta_{base}$=[1.0, 1.0, 1.0, 1.0, 0.0]) | 37.0 / 38.3 | 41.9 / 37.2 |
| SNP++ (Removal x2; $\beta_{base}$=[1.0, 1.0, 1.0, 0.0, 0.0]) | 29.8 / 36.9 | 33.4 / 32.9 |

| | Avg accuracy after Task 2 | Avg accuracy after Task 3 |
| --- | --- | --- |
| Interpolation | | |
| SNP++ ($\beta_{int}$=0.1) | 26.1 / 38.8 | 39.4 / 33.6 |
| SNP++ ($\beta_{int}$=0.5) | 34.7 / 38.1 | 37.2 / 37.2 |
| SNP++ ($\beta_{int}$=1.0) | 36.3 / 41.0 | 36.5 / 34.9 |
| Mode switching | | |
| SNP++ (S=1000, K=5) | 37.0 / 36.6 | 38.6 / 35.8 |
| SNP++ (S=1000, K=10) | 37.7 / 36.5 | 34.1 / 34.9 |
| SNP++ (S=10,000, K=5) | 37.1 / 36.8 | 34.7 / 36.2 |
| SNP++ (S=10,000, K=10) | 37.0 / 36.6 | 39.3 / 37.2 |

existing base parameter is closest to the new task, and use this is the target base parameter. Then we can update the meta parameters such that, while the meta parameters drift and other non-target base parameter drift is minimized, the target base parameter is being updated towards the new task while performing well on its prior tasks.

For mode switching, the parameter space has many functionally-diverse modes that we may wish to use and replace an existing subnetwork in-place. For example, we could replace a task's base parameter with an adversarially-robust parameter (e.g. from adversarial training), or a backdoor-robust parameter (e.g from backdoor adversarial training), or domain-robust parameters, etc. Rather than using the task's original training set to locate this mode, an alternative approach would actively sample the parameter space, and update the meta parameters and regularize the drift of the replaced mode's base parameter such that the new base paramater is computed with respect to the specific task. While it is possible to actively sample the base parameters iteratively to identify the ideal base parameter mode, it poses a risk that the target mode may cause the meta parameter to drift beyond the subspace radius. Thus, for our evaluation of the identification of a sharpness-aware mode (for low-loss basins using SAM (Foret et al., 2021)), we actively sample meta parameters graduating from within the radius to outside of the radius, and for each sampled meta parameter we compute the base parameter, and evaluate whether it satisfies the mode's evaluation condition (e.g. flat basin).

## 6 EXPERIMENTS

### 6.1 METHODOLOGY

**Model.** We evaluate with CLIP (Radford et al., 2021), the standard vision-language model, specifically the pre-trained initialization of the ResNet-50 variant. For training/fine-tuning on a new task, we retain the Adam optimizer, decoupled weight decay regularization, temperature clipping, and a batch size of 32. From the pre-trained initialization, we train CLIP on MSCOCO for 50 epochs until both loss convergence and verification that zero/few-shot performance across the datasets is weakened. We fine-tune for 10 epochs and also validate loss convergence.

**Adaptation baselines.** Fine-tuning (Single Task Learning) is a baseline in online/continual learning where the model sequentially learns each incoming task without any forgetting mitigation. Joint Training (Multi Task Learning) is a baseline where the model trains on all future tasks. We do not include the base task (MSCOCO), and evaluate when there are at least 2 tasks (Task 3).

We evaluate against 5 baseline adaptation methods, including 3 general-purpose continual learning strategies, and 2 large-model-specific adaptation strategies (that have also been evaluated on CLIP). Elastic Weight Consolidation (EWC) (Kirkpatrick et al., 2017) uses weight regularization to retain weights from previous tasks. The regularization strength for weight penalty $\lambda$ is $1,000$. Gradient Projection Memory (GPM) (Saha et al., 2021) learns new tasks by performing gradient steps in the orthogonal direction to the gradient subspaces that are important to the past tasks. We use a 0.01 learning rate. BatchEnsemble (Wen et al., 2020) uses a base network (slow weights) and stores separate parameters (fast weights) to compute the parameters per ensemble, thus $N$ ensembles do not require $N$ sets of parameters. Each ensemble member is responsible for one task. We retain -0.5 random sign init for fast weights and 0.5 fast weights learning rate multiplier.

CLIP-Adapter (Gao et al., 2021) fine-tunes with feature adapters, specifically an additional bottleneck layer to learn new features and perform residual-style feature blending with the original pre-trained features. In-line with the implementation for CLIP, we fine tune the visual adapter. PAINT (Ilharco et al., 2022) is another vision-language model adaptation technique. It is a patching method that interpolates between parameters before fine-tuning and parameters after fine-tuning on a task to be patched. We implement sequential patching, where we iteratively repeat the patching procedure on each new task, and pick the mixing coefficient that optimizes average accuracy on the held-out validation sets from the supported and patching tasks. All baselines are trained with the CLIP model.

**Zero-shot.** Unlike visual systems trained on a fixed set of discrete labels, Radford et al. (2021) proliferates a paradigm in learning to align images with texts in an open-vocabulary setting. For zero-shot classification, class labels are converted to sentences using prompt templates, and the model computes text embeddings. The model computes the image embedding, and an image-text (cosine) similarity score is computed between the image embedding and each class' text embeddings

for strongest alignment. Similarity of embeddings per class, scaled by a temperature parameter, is normalized into a probability distribution via softmax. The class having highest similarity with the image is the predicted one. We perform zero-shot evaluation with the altered CLIP-Adapter model and with the task-indexed BatchEnsemble model. For other methods, the model architecture and parameters are available for directly applying this zero-shot evaluation procedure.

**Few-shot.** For a given task, an N-way-K-shot support set (N classes, K samples per class) is provided for inference. Evaluation is performed on the query set. The meta learner computes the base parameters with respect to the support set. Specific to gradient-based meta learners, including MAML and SNP(++), we compute the gradient of the support set with respect to model parameters, update the model parameters, then evaluate on the query set. For standard CLIP and the other methods, we use nearest-neighbor classification. We first compute the mean image features per class in the support set, then measure the (cosine) distance between them and the image features for a given query set image. Based on the nearest class mean, the closest class' mean image features is the predicted class.

## 6.2 MAINTAINING ZERO/FEW-SHOT CAPABILITIES

We tabulate our proposed method in comparison to baselines in Table 7, and proposed method in comparison to different configurations in Table 8. Transferability between tasks plays an important role. From task 1 to 3, we find that positive backward transfer exists across all baselines, and that some datasets have improved zero/few-shot performance with task shift. Furthermore, we find that the gradual removal of subnetworks with SNP++ may worsen performance. The removal of subnetworks is motivated by alleviating interference between task-specific representations between two tasks; in this case, it appears that the attempt at removal overwrites the transferable representations. Further sub-procedures that identify the optimal subnetworks to remove based on a transfer-interference trade-off can improve the utility of subnetwork removal, especially in a setting with many tasks. Adaptation techniques specialized for large models (CLIP in particular) outperform general-purpose continual learning methods. For large models, regularization-based methods that do not require task indexing or separate context vectors can perform competitively to non-regularization-based methods.

Our proposed adaptation method, SNP and SNP++, outperforms existing baselines. It can consistently retain low negative backward transfer, fulfilling the objective of minimizing loss of zero/few-shot performance with task shift. It performs comparably in maximizing positive backward transfer. In terms of balancing between positive and negative backward transfer, SNP and SNP++ strikes the optimal balance, attaining the highest average accuracy. We find that our proposed method works better with the memory buffer that regularizes the base parameter drift. Though we are not storing trajectories or large replay buffers (only storing one support set instance), pure-regularization SNP can also perform in a stable fashion. Different hyperparameters of $\beta_{base}$ and $\beta_{meta}$ tend to retain similar performances, and no major loss in accuracy is observed. We do note that setting the $\beta_{meta}$ too low can worsen performance, particularly in comparison to non-SNP baselines. This may occur where the drift of the meta parameters is under-regulated, and regulating base parameter drift is insufficient and acts as a second-order regularizer of the meta parameter drift. We find that combining subnetworks and interpolating between them underperforms SNP and additive SNP++. Subnetwork addition/removal would manipulate the number of base parameters, but result in first-order interpolation between the unchanged meta parameters and new meta parameters with the modified subnetwork set. Thus, interpolating between subnetworks results in second-order interpolation, and the error with respect to each task accumulates when the meta parameters interpolate. Given a large number of tasks and lower model capacity, second-order interpolation offers an efficient subnetwork manipulation.

## 7 CONCLUSION

By projecting a model's subnetworks onto the same equidimensional parameter space as the (meta) parameters, we can edit the representations encoded in the network, including adding, removing, combining, and switching subnetworks. We apply this paradigm to achieve superior online/continual learning performance in retaining seen and zero/few-shot accuracy on prior and subsequent tasks. Not only does our method scale to large models, it also lays the foundation for further network editing applications, such as subnetwork removal for privacy (e.g. machine unlearning), or subnetwork addition for distributional robustness (e.g. adding distributionally-varied samples for fairness or adversarial robustness).

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
