# OpenReview forum: "Projected Subnetworks Scale Adaptation"
_ICLR.cc/2024/Conference — Submitted to ICLR 2024_

### Official Review · Reviewer_z8z7 · 2023-10-30

**Soundness:** 2 fair
**Presentation:** 1 poor
**Contribution:** 2 fair
**Rating:** 3
**Confidence:** 4

**Summary:**

The research delved into the challenges large models face when trying to generalize in zero-shot and few-shot situations during continual learning. Unlike previous studies that mainly focused on performance memory within domain-specific tasks, this study introduces a technique to maintain performance for both in-domain and out-of-domain tasks.

**Strengths:**

SNP surpasses other benchmarks in multiple tasks, including zero-shot and few-shot learning.

**Weaknesses:**

(1) The authors' approach to the continual learning problem is ambiguous. I'm uncertain about the specific issue they're addressing, and I'm concerned that the authors might also lack clarity on the subject.

(2) The objective function defines base parameters using all the tasks in the set. Yet, this conflicts with the fact that tasks are presented to the learner sequentially. How can a base learner be defined using all the data when it only becomes accessible at the end of its operation? This inconsistency suggests a fundamental flaw in the framework.

(3) It seems that task groupings are determined using data from all tasks in Sec. 3.1. This suggests the experimental design relies on complete task data, making it questionable. A sound experimental design should use randomized trials or employ specific methodologies to assess and counteract biases from non-random setups. However, it appears the authors recognize biases in the data and use them to shape their experimental approach. This methodology is problematic and lacks validity.

(4) Regarding the presentation, the placement of tables separate from their explanatory text makes them challenging to follow. A suggestion for improved clarity would be to bold the top-performing metrics in each table corresponding to specific tasks.

**Questions:**

(1) Does the meta-learning algorithm employ a second-order gradient update?

(2) Given that gradient-based meta-learning might elevate model complexity, it's important to address how SNP impacts factors like model size, computational requirements, and overall intricacy.

(3) Would the authors be able to delve deeper into this process? How does it counteract the rise in base parameter error when the shift in meta parameters surpasses the allowed radius?

---

### Official Review · Reviewer_oCJY · 2023-10-30

**Soundness:** 3 good
**Presentation:** 1 poor
**Contribution:** 2 fair
**Rating:** 3
**Confidence:** 4

**Summary:**

This paper proposed a meta-learning objective to train a meta model with operates well across a sequence of tasks it is finetuned on (though unfortunately, exact details of the proposed approach are quite unclear and obfuscated). When deployed across a large range of dataset-adaptation-sequences, the proposed subnetwork projection showcases promising performance, particular for task sequences large than two.

**Strengths:**

* The experimental results are quite convincing once the third task is reached. Before that, performance closely resembles that of PAINT-style adaptation.
* The suite of adaptation experiments is quite expansive and covers a dozen of common evaluation benchmarks.

**Weaknesses:**

The primary issue with this paper comes from its presentation and writing, which is incredibly difficult to parse, lacks detail and often motivation and descriptions for experiments and components. In particular:

* First, and most importantly, the proposed SNP/SNP++ approach is never really formalized, with the base-text just references large, ad-hoc algorithm chunks without providing any actionable information on the method details, and what actually happens. Similarly, subnetworks, as used and target in this work, are never actively defined.
As a result, it is very difficult (near impossible) to understand what the method is exactly supposed to do. It would be important for the authors to provide some better context here.

* The paper effectively has two separate experiment sections. While it's clear that the first one serves to motivate the method, it lacks a lot of context that makes it difficult to understand some of the experimental specifics without constantly jumpting back and forth. In particular:
	* For 3.1, it would be great to first lead with a quick summary of what this section aims to investigate. Generally, it remains somewhat unclear what the exact point of this section is. Is it just to highlight that models finetuned on different datasets exhibit differences with respect to each other? And if so, a more detailed investigation for example relating this to semantic relations between datasets would be much more insightful, instead of just highlight some arbitrary numerical grouping.

	* Section 3.2 starts to talk about subnetworks, without those being introduced beforehand. In general, it is again difficult to understand the purpose of this section and the experiments conducted in it.


* The discussion of related works is also quite lackluster, and misses a relevant discussion of both recent works looking at interpolation-based adaptation of foundation models for CL (e.g. [1]), as well as the large body of research investigating continual learning and adaptation through architectural restructuring (e.g. [2-4]). In general, the overall discussion of continual learning works could certainly be improved.

* This work often leverages metalearning as a driving motivator. But how crucial is a Meta-Learning approach in light of works such as [7,8] which highlight limited importance of a full meta-learning objective for few-shot transfer?


__Smaller issues regarding the general writing:__

* The abstract is very short, and could benefit from some additional notes on the subnetwork selection and its motivation, alongside some further details about the experiments.

* The exact values for Tab. 1 aren't crucial for the readers understanding, and the table would be much better replaced with a corresponding heatmap. Same goes for Table 2.

* Tables are often placed awkwardly far from the first point of reference (e.g. Tab.7 & 8). In general, they are so dense that they become difficult to parse.

* It would be great to elaborate on the distance function dist(.,.) as soon as it is introduced (p.3)

* Algorithm 1 is never even explicitly referenced (and Algorithm 2 not in order), only the correspondingly defined functions are used in subsequent algorithms. It would be great to provide crossreferences here.

* In general, standard deviations would be great for all reported numbers.



__Related Works__
* [1] "Momentum-based Weight Interpolation of Strong Zero-Shot Models for Continual Learning", Stojanovski et al. 2022
* [2] "PackNet: Adding multiple tasks to a single network by iterative pruning", Mallya et al. 2018
* [3] "Progress & Compress: A scalable framework for continual learning", Schwarz et al. 2018
* [4] "Compacting, Picking and Growing for Unforgetting Continual Learning", Hung et al. 2019
* [5] "An empirical investigation of the role of pre-training in lifelong learning", Mehta et al. 2022
* [6] "Effect of scale on catastrophic forgetting in neural networks", Ramasesh et al. 2022
* [7] "A Baseline for Few-Shot Image Classification", Dhillon et al. 2019
* [8] "Exploring Simple Meta-Learning for Few-Shot Learning", Chen et al. 2020

**Questions:**

This paper resides in an awkward spot, where the experiment are convincing, and the high-level motivation does appear sensible, but there the writing and structuring, alongside the severe lack of detailing, makes it impossible to replicate and near impossible to understand the exact details and motivation of the proposed approach.
Consequently, I am currently advocating for rejection, but would be willing to update my score if the authors could elaborate on the issues noted above - in particular providing a clear definition of subnetworks used in this paper, and a detailed explanation of each step that goes into the proposed subnetwork projection.

---

### Official Review · Reviewer_EH1D · 2023-10-30

**Soundness:** 1 poor
**Presentation:** 1 poor
**Contribution:** 2 fair
**Rating:** 1
**Confidence:** 4

**Summary:**

The authors present a method that employs gradient-based meta-learning to adapt models to specific tasks while attempting to preserve few-shot and zero-shot capabilities on other (seen or unseen) tasks. Essentially, the authors penalize the drift (distance) between different sets of base and meta parameters created through a range of interpolations in parameter space.

**Strengths:**

-	The authors present some interesting ideas and insights on the behavior of parameter drift in meta-learning scenarios and how to potentially use distance regularization in this space
-	They also try to improve clarity by providing their algorithms in pseudo code;

While there definitely are some interesting points to this method, there is significant further work that needs to be done by the authors on their manuscript;
&#8594; Please see Weaknesses section.

**Weaknesses:**

While I can see that the authors are following a potentially interesting idea and I am aware that some of the ‘issues’ might be due to language difficulties, the current state of the manuscript does in my opinion not warrant publication due to several severe issues; Please see below.

---
### Clarity of story line, goal and contributions:
The paper is unfortunately rather hard to read and follow; Even reading through the abstract is somewhat confusing and it doesn’t entirely become clear what the authors are going to demonstrate in this work.
&#8594;  I’d therefore like to encourage the authors to reformulate the abstract and include some more details, underlying intuitions of their approach as well as a glimpse towards the main contributions.
&#8594; The same applies to the introduction and many further parts of the paper; I am aware that this might be due to language difficulties, but I’d highly recommend the authors to go through their manuscript again and improve the clarity of their argument(s).

Some examples:
Paragraph 1 start with distribution shifts (time, person (?), environment) – then moves on to few-shot learning with base learners and meta learning; then scaling of models and data, then suddenly to online settings and ‘prior’ continual learning methods; before finally jumping to disentanglement in early paragraph 2.
This doesn’t create a conclusive story line to follow as there are no real connections between these aspects pointed out, neither does it become clear how these things relate to the work at hand -- therefore the structure would significantly benefit from further (re)work.

---
### Insufficient placement in context of related work:
The authors simply state a variety of other works without relating them to their own contributions. It is entirely unclear from this section how the authors’ own work is placed in the context of these works, why they are related and to which extend – and how their own work differs.

---
### Lacking quality & clarity of method description:
Generally, the authors tend to simply state what is done – but largely fail to provide insight as to WHY certain steps are taken, what the underlying motivation is, what we should expect, what they want to show, etc.;
Some examples (not comprehensive):
- The authors state in Sec. 3.1 that the pre-trained model **already performs well across all datasets**; yet, they train on MSCOCO to **overfit** to the point where the model **does no longer perform well** on the other sets; In other words, the authors **create their own problem that they want to then solve**. While this can be done, I am entirely **missing the intuition/justification that is underlying this approach**.
- Why don’t the authors start from the ‘well-performing’ pretrained backbone and try to further improve on the small datasets? Why don’t the authors choose to start from a backbone that’s solely trained on MSCOCO to begin with (which would be more alike other continual learning works, i.e. iteratively learning new skills).

- While I can see the idea of comparing “functional distances” based on model performance in Table 2, what is the underlying justification that the cosine distance between the flattened parameter sets directly represents the ‘distance’ between these datasets (Table 1)? What happens if you simply choose another random seed for the SAME dataset – wouldn’t you also end up with a different set of parameters that simply converged to a different local minimum, and all distances might therefore change?
- Section 3.1 would equally benefit from some more details regarding underlying motivation, intuition etc. – as would Section 3.2.
- Section 3.2 suddenly splits ONE dataset into several tasks (Split-MSCOCO), whereas up to this point, different datasets were considered – again, it would help to provide insight as to WHY this experiment is performed.
- Section 3.3. describes investigations on the drift of meta parameters that can be tolerated and states at the end that the authors are interested in “maximizing the radius of the parameter subspace […]” – it would however again significantly help the reader WHAT the underlying intuition and purpose of the experiments is.

---
### Some inconsistencies throughout:
E.g.:
- Section 3.2 states that “we can project the task-specific representations… within the meta parameters to the base parameter space”, but the authors already stated a few sentences earlier that “meta-parameters and base parameters reside in the same parameter space”. This contradicts itself since there is no need to project anything if both already live in the same space.

---
### Unstated implications & Limitations:
Some examples:
- Algorithm 3 shows that the base parameters for all tasks have to be stored (line 6.); Further, an additional set of interpolated base parameters is stored for each task throughout the interpolation procedure (line 15). &#8594; Comments on the additional required memory, especially for large models and many tasks feels necessary & helpful!
- Algorithms 3 loops through all tasks to determine base parameters, then through the tasks again to compute task losses, followed by an additional nested loop over all random interpolation params ‘S’ (here 1000) with a set of interpolation constants within; &#8594;  Commenting on the train complexity in terms of runtime would be helpful as well.

&#8594;  Note that discussing this becomes imperative especially because the authors claim in the conclusion that their method “scales to large models”

**Questions:**

Please refer to the ‘Weaknesses section’.

I highly encourage the authors to rework their manuscript and pay special attention to **providing insights and the underlying motivation** as to **WHY** steps are taken, what the purpose is, what we expect to see, etc.

---

### Official Review · Reviewer_bDN1 · 2023-10-31

**Soundness:** 1 poor
**Presentation:** 2 fair
**Contribution:** 2 fair
**Rating:** 1
**Confidence:** 4

**Summary:**

The paper tackles the problem of learning with limited labels in the context of continual learning. It develops the approach based on subnetworks whose parameters are represented in the same space as base learner parameters, resulting in learning composable and editable network representations. I think overall this is an interesting idea, akin to hypernetworks. However, the paper does not demonstrate that the derived representations indeed possess the claimed properties. Moreover, the presentation of the paper needs significant improvement, something that would require a major rewrite of the entire manuscript. Although I see value in the idea and empirical results, the revision that is necessary to make the paper fit quality criteria of a conference such as ICLR, in my opinion, is well beyond what is typically achievable during a conference paper revision. Therefore I recommend reject.

**Strengths:**

- interesting and exciting topic
- interesting results comparing CLIP and MAML
- extensive empirical results

**Weaknesses:**

- I believe there are some typos in equations, or otherwise the descriptions are not clear
- Writing clarity, structure and flow need to be improved
- the setup seems compromised as the accuracy in unseen datasets is used to guide model training
- the paper is written as a technical report describing what was done. for a scientific paper, I would expect a lot more high-level
- motivatioon of design and theory supporting the algorithm development.
- The paper claims that the subnetwork representations can be added, removed, edited, etc. In other words, composed. However, this is not backed up by empirical evidence.

**Questions:**

- The paper makes references to LLMs in the intro, but no large scale experiments are provided. This may feel misleading.
- I believe some of the relevant literature is missing. For example, Progressive Neural Networks https://arxiv.org/abs/1606.04671 achieve the addition of new tasks without forgetting by adding extra parallel branches
- I believe that the equation in line 135 is incorrect. Shouldn't it be $\theta_{base, t-1}$ on RHS?
- Section 3.1 jumps out of nowhere. I suggest that the Algorithm 1 is deleted, because it is trivial and is not referenced anywhere anyways. Instead, you could spend this space to do a proper intro in Section 3 and transitions. Otherwise it is really hard to understand where this whole thing is going.
- I believe 3.1. really belongs to the results section as it describes datasets. It seems irrelevant in the context of method description.
- Tables dumped on page 2 with no explanations of their relevance prevent effective digestion of the manuscript materials
- "From the pre-trained initialization, we train on the first task of MSCOCO for 50 epochs, until the accuracy is high on MSCOCO and low for all the unseen datasets that the pre-trained model performed well on". It sounds like the performance of the unseen datasets is tracked while training the model. This sounds to me like an information leak from the unseen datasets to the model that is not supposed to use them at train time in any way.
- In table 1, can you mark the 3 identified groups of dataset by color and also implement some hierarchical clustering to put similar datasets together? Otherwise, it is very hard to see the point. Can you move the table on the same page with Section 3.1 where it is referenced?
- Would it be possible to retain only Table 2 in the main text (table 1 goes to appendix). What is the value of having both of them in the main body? Do they provide similar or different insights? This has to be discussed somehow. Right now I see that only Table 2 is really used to create dataset groupings at the end of Section 3.1. Again, I emphasize that it feels that all of this material better belongs to experimental section, unless authors properly motivate why they want to put it in the method section, which they clearly do not have space to do right now, because they put too many tables and algorithms in the paper. This way, it is very hard for me to see the forest behind the trees.
- "To retain the same scale of model and data, we use the same CLIP architecture and capacity" Neither the architecture was described, nor its capacity was quantified. Again, Section 3.2 reads as if it should be part of the experiments section.

---

### Meta-Review · Area_Chair_K48z · 2023-12-07

**Metareview:**

This paper investigates an effective adaptation of pre-trained models without hurting zero-shot performance. All knowledgeable reviewers recommend rejection due to the quality of the presentation, unclear justification of goal and approach, and concerns about experiments. The authors did not respond in a rebuttal. AC agrees with the reviewers' decision and recommends rejection.

**Justification For Why Not Higher Score:**

N/A

**Justification For Why Not Lower Score:**

N/A

---

### Decision · Program_Chairs · 2024-01-16

Reject